# Batched Coarse Ranking in Multi-Armed Bandits

**Nikolai Karpov**
Department of Computer Science
Indiana University
Bloomington, IN 47405
nkarpov@iu.edu

**Qin Zhang**
Department of Computer Science
Indiana University
Bloomington, IN 47405
qzhangcs@indiana.edu

## Abstract

We study the problem of coarse ranking in the multi-armed bandits (MAB) setting, where we have a set of arms each of which is associated with an unknown distribution. The task is to partition the arms into clusters of predefined sizes, such that the mean of any arm in the $i$-th cluster is larger than that of any arm in the $j$-th cluster for any $j > i$. Coarse ranking generalizes a number of basic problems in MAB (e.g., best arm identification) and has many real-world applications. We initiate the study of the problem in the batched model where we can only have a small number of policy changes. We study both the fixed budget and fixed confidence variants in MAB, and propose algorithms and prove impossibility results which together give almost tight tradeoffs between the total number of arms pulls and the number of policy changes. We have tested our algorithms in both real and synthetic data; our experimental results have demonstrated the efficiency of the proposed methods.

## 1 Introduction

We study the *coarse ranking* problem in multi-armed bandits (MAB). In this problem we are given a set of $n$ arms, each of which is associated with an unknown distribution $\mathcal{D}_i$ on support $[0, 1]$; let $\theta_i$ be the (unknown) mean of $\mathcal{D}_i$. We are also given a vector $\mathbf{m} = (m_0, m_1, \ldots, m_k)$, where $k$ is a predefined parameter and $0 = m_0 < m_1 < \ldots < m_k = n$. Our goal is to partition the $n$ arms into $k$ clusters such that the means of arms between those clusters are sorted. That is, the first cluster contains the $(m_1 - m_0)$ arms with the largest means, the second cluster contains the $(m_2 - m_1)$ arms with the largest means in the remaining arms, and so on. We perform the clustering by pulling the $n$ arms; by pulling the $i$-th arm we mean to obtain a sample from the distribution $\mathcal{D}_i$. The goal is to learn the correct clustering using as few pulls as possible.

The coarse ranking problem naturally arises in many real-world applications, including recommendation systems for movies and books [3], peer grading for ranking students in massive open online courses (MOOC) [33, 35], and ranking experts in paid crowdsourcing platforms. In these applications we try to partition entities (movies, books, experts, etc.) into clusters and give different ratings/credits to different clusters. The problem also generalizes a number of basic problems in MAB including *best arm identification* ($k = 2$, $m_1 = 1$), *top-m arm identifications* ($k = 2$, $m_1 = m$), and *arm sorting* ($k = n$, $m_i = i$ for $1 \le i \le n$).

The coarse ranking problem is closely related to the problem of *ranking from pairwise comparison*, where we want to sort $n$ elements by *noisy* pairwise comparisons. For each comparison on the pair of elements $(i, j)$, the algorithm gets a feedback that the $i$-th element is better than the $j$-th element with (unknown) probability $p_{i,j}$ (or equivalently, the $j$-th element is better than the $i$-th element with probability $p_{j,i} = 1 - p_{i,j}$). The order of the $n$ elements can be defined using the Borda score $\theta_i \triangleq \frac{1}{n-1} \sum_{j \ne i} p_{i,j}$ for $i \in \{1, \ldots, n\}$, which corresponds to the probability that the $i$-th element is better than an element chosen uniformly at random among the other $(n - 1)$ elements. This reduction,

named Borda reduction [21], allows us to consider the problem of ranking from pairwise comparison in the MAB setting.

In this paper we study the coarse ranking problem in the *batched* model: Instead of pulling the arms one by one adaptively, we pull them in batches (or, *rounds*). The set of arms to be pulled in each round needs to be determined at the beginning of the round. Our goal is to characterize the tradeoffs between the total number of pulls and the number of rounds used in the learning process. A small number of rounds corresponds to a small number of learning policy changes, which is desirable in practice since it enables parallelism and reduces the total running time of the learning process. For example, in crowdsourcing, there could be a significant waiting time to get the answer from the crowd. In medical trials, it can take a few days to observe the effects of the drugs. In these scenarios, a fully adaptive policy simply does *not* work.

**Our Contributions.** In this paper we give the first study of the coarse ranking problem in the batched model, and characterize the tradeoffs between the total number of arm pulls and the number of rounds of the learning process.

We study two variants of the problem which are standard in the literature of MAB. The first is called *fixed budget* [15, 28], where the goal is to minimize the error probability given a fixed amount of pulls. The second is called *fixed confidence* [6], where the goal is to minimize the number of pulls to achieve a predefined error probability. Before presenting our results, we first introduce the following concepts. Let $[n] = \{1, 2, \ldots, n\}$. Assume all logarithms are of base 2.

**Definition 1** (Mean Gap). *Given an input set of arms $I = \{1, 2, \ldots, n\}$ and a parameter vector $\mathbf{m} = (m_0, m_1, \ldots, m_k)$ with $0 = m_0 < m_1 < \ldots < m_k = n$, let $\theta_{[i]}(I)$ be the $i$-th largest mean of arms in $I$. For convenience define $\theta_{[0]}(I) = +\infty$ and $\theta_{[n+1]}(I) = -\infty$. For any $i \in I$, let $j \in [k]$ be the index such that $\theta_{[m_j]}(I) \leq \theta_i \leq \theta_{[m_{j-1}+1]}(I)$. Define the gap for the $i$-th arm to be*

$$\Delta_i^{\langle \mathbf{m} \rangle}(I) \triangleq \min\{\theta_{[m_{j-1}]}(I) - \theta_i, \theta_i - \theta_{[m_j+1]}(I)\}.$$

We propose the following quantity to characterize the complexity of an input instance for the coarse ranking problem.

**Definition 2** (Instance Complexity). *Given an input instance $(I, \mathbf{m})$ of the coarse ranking problem, define*

$$H^{\langle \mathbf{m} \rangle}(I) \triangleq \sum_{i \in I} \left( \Delta_i^{\langle \mathbf{m} \rangle}(I) \right)^{-2}.$$

We note that this definition of instance complexity for coarse ranking generalizes the one defined for best arm identification [4] and that for top-$m$ arm identifications [6].

We have the following property for the instance complexity $H^{\langle \mathbf{m} \rangle}(I)$. The proof of the proposition is deferred to Appendix B.1 (see the supplementary material).

**Proposition 1.** *Let $\pi : \{1, \ldots, n\} \to I$ be the bijection such that $\Delta_{\pi(1)}^{\langle \mathbf{m} \rangle}(I) \leq \ldots \leq \Delta_{\pi(n)}^{\langle \mathbf{m} \rangle}(I)$. We have $\max_{i \in I} \left\{ i \cdot \left( \Delta_{\pi(i)}^{\langle \mathbf{m} \rangle}(I) \right)^{-2} \right\} \leq H^{\langle \mathbf{m} \rangle}(I) \leq \log(2n) \cdot \max_{i \in I} \left\{ i \cdot \left( \Delta_{\pi(i)}^{\langle \mathbf{m} \rangle}(I) \right)^{-2} \right\}.$*

The main results of this paper are summarized as follows:

1. In the fixed budget case with pull budget $T$, we give the first algorithm for the batched coarse ranking problem that uses $R$ rounds and succeeds with probability $1 - \exp\left(-\tilde{\Omega}\left(\frac{T}{n^{1/R} \cdot R \cdot H^{\langle \mathbf{m} \rangle}(I)}\right)\right)$, where '$\tilde{\Omega}(\cdot)$' hides logarithmic factors which will be spelled out in Section 2. In particular, we show that $\log n$ rounds and $\tilde{O}(H^{\langle \mathbf{m} \rangle}(I))$ pulls are enough to achieve a 0.99 success probability. We also complement this upper bound with an almost matching lower bound (Section 4). We note that the coarse ranking problem has not been studied in the fixed budget setting even for fully adaptive algorithms.

2. In the fixed confidence case with error probability $\delta$, we give the first algorithm which solves the batched coarse ranking problem successfully with probability at least $(1 - \delta)$ using

$$O\left(\log\max_{i\in I}\left\{1/\Delta_i^{\langle \mathbf{m}\rangle}(I)\right\}\right)$$ rounds and $\tilde{O}\left(H^{\langle\mathbf{m}\rangle}(I)\right)$ arm pulls (Section 3). We again complement our upper bound with an almost matching lower bound (Section 4).

**Related Work.** The coarse ranking problem has recently been studied in the MAB model [30], but the algorithm proposed in [30] is fully adaptive (UCB-based) and is thus not applicable to the batched setting. In general, it seems difficult to adapt UCB-type algorithms to the batched setting, since the arm pulls in UCB-type algorithms are inherently sequential. The complexity measure (instance complexity) in [30] is different from ours, and thus the sample complexities of the two algorithms are not directly comparable. The problem is also investigated in noisy comparison model [19, 18, 34] where the algorithms are again inherently adaptive. As mentioned, the coarse ranking problem is a generalization of several basic problems in the MAB setting including best arm identification and top-$m$ arm identifications, which have been studied extensively in the literature in the fully adaptive model [14, 26, 4, 27, 15, 22, 6, 28, 20, 32, 9, 31, 25, 10]. Best arm and top-$m$ arms identifications have also been studied in the noisy comparison model in the non-adaptive or fully adaptive settings [7, 36, 23, 18, 11, 2].

In recent years, a number of fundamental problems in MAB, reinforcement learning and online learning have been studied in the batched model, including best/top-$k$ arm identifications [25, 1, 24], regret minimization [32, 16, 13], $Q$-learning [5], convex optimization [12], and online learning [8]. The best arm identification is also studied in a model named *collaborative learning* [20, 37], where multiple agents try to learn a target function together by pulling the arms in rounds. As we shall explain in more detail in Section 4 and Appendix B.4, algorithms designed for the batched model can be easily translated to *non-adaptive* algorithms in the collaborative learning model.

**Notations.** Table 1 summarizes a set of notations that we will use in the rest of this paper.

| $n$ | number of arms in the input |
|---|---|
| $T$ | time budget |
| $\theta_i$ | mean of the $i$-th arm |
| $\theta_{[i]}(I)$ | the $i$-th largest mean among arms in $I$ |
| $\mathbf{m}$ | the cluster boundary vector |
| $\Delta_i^{\langle\mathbf{m}\rangle}(I)$ | mean gap of the $i$-th arm of input $I$; see Definition 1 |
| $H^{\langle\mathbf{m}\rangle}(I)$ | instance complexity of input $I$; see Definition 2 |

Table 1: Summary of Notations

## 2 The Fixed Budget Case

We present our algorithm for the fixed budget case, named SRank, in Algorithm 1. The algorithm proceeds in $R$ rounds. In each round $r$ ($0 \leq r \leq R-1$) we pull each arm for a number of times and record their empirical means (Line 5-7). We then identify the set of arms that have the largest empirical mean gaps (Line 8-10) and assign them to the respective clusters (Line 11-15). We then work with the remaining arms in the next round.

Intuitively, Algorithm 1 always maintains a set of arm $I_r$ that remain to be clustered. Arms in $I \setminus I_r$ have already been partitioned into clusters $C_1^{(r)}, \ldots, C_k^{(r)}$. Now, if we cluster $I_r$ according to the vector $\mathbf{m}^{(r)}$ to $D_1^{(r)}, \ldots, D_k^{(r)}$, then $C_j^{(r)} \cup D_j^{(r)}$ forms the $j$-th cluster in the desired clustering of $I$ (determined by the input vector $\mathbf{m}$). At each iteration, the algorithm identifies a subset of arms $E_r$ and extends clusters $C_j^{(r)}$ to $C_j^{(r+1)}$ by adding each arm from $E_r$ to the cluster to which the arm should belong to.

Algorithm 1 is inspired by the *successive accepts and rejects (*SAR*)* algorithm [6], but it is non-trivial to extend SAR which was designed for top-$m$ arm identifications to coarse ranking. In particular, we have to redesign the algorithm in order to achieve the newly proposed instance complexity (Definition 2) for coarse ranking; the subsequent analysis also needs significant new ideas. Moreover, we need to augment the algorithm to handle batched pulls.

---

**Algorithm 1:** $\mathtt{SRank}(I, \mathbf{m}, T, R)$

---

**Input:** a set of $n$ arms $I$, cluster boundary vector $\mathbf{m} = (m_0, m_1, \ldots, m_k)$ with
      $0 = m_0 < m_1 < \ldots < m_k = n$, and time horizon $T$
**Output:** the coarse ranking ($k$-clustering) of arms in $I$

1   Initialize $I_0 \leftarrow I$, $\mathbf{m}^{(0)} \leftarrow \mathbf{m}$, and for $j \in \{1, \ldots, k\}$ set $C_j^{(0)} \leftarrow \emptyset$ ;

2   set $T_0 \leftarrow 0$, $T_r \leftarrow \left\lfloor \frac{n^{r/R} \cdot T}{n^{1 + 1/R} \cdot R} \right\rfloor$ for $r = 1, \ldots, R$ ;

3   $n_r \leftarrow \left\lfloor \frac{n}{n^{r/R}} \right\rfloor$ for $r = 0, \ldots, R - 1$ and $n_R \leftarrow 0$ ;

4   **for** $r = 0, 1, \ldots, R - 1$ **do**

5      pull each arm in $I_r$ for $T_{r+1} - T_r$ times ;

6      let $\hat{\theta}_i^{(r)}$ for $i \in I_r$ be the empirical mean of the $i$-th arm after $T_r$ pulls ;

7      let $\sigma_r : \{1, \ldots, n_r\} \to I_r$ be the bijection such that $\hat{\theta}_{\sigma_r(1)}^{(r)} \geq \hat{\theta}_{\sigma_r(2)}^{(r)} \geq \ldots \geq \hat{\theta}_{\sigma_r(n_r)}^{(r)}$, and for
       convenience $\hat{\theta}_{\sigma_r(0)}^{(r)} = +\infty$, $\hat{\theta}_{\sigma_r(n_r+1)}^{(r)} = -\infty$ ;

8      let $\{\hat{C}_1^{(r)}, \ldots, \hat{C}_k^{(r)}\}$ be the partition of $I_r$ into $k$ parts where $\hat{C}_j^{(r)} \leftarrow \{\sigma_r(i)\}_{i = m_{j-1}^{(r)} + 1}^{m_j^{(r)}}$ ;

9      for $j \in \{1, \ldots, k\}$ and $i \in \hat{C}_j^{(r)}$ define the empirical gap as
$$\Delta_i^{(r)} \leftarrow \min\left\{ \hat{\theta}_{\sigma_r\left(m_{j-1}^{(r)}\right)}^{(r)} - \hat{\theta}_i^{(r)}, \hat{\theta}_i^{(r)} - \hat{\theta}_{\sigma_r\left(m_j^{(r)}+1\right)}^{(r)} \right\} ;$$

10     let $E_r \subset I_r$ be the set of $(n_r - n_{r+1})$ arms with largest empirical gaps $\Delta_i^{(r)}$ ;

11     **for** $j = 1, \ldots, k$ **do**

12       $C_j^{(r+1)} \leftarrow C_j^{(r)} \cup \left( \hat{C}_j^{(r)} \cap E_r \right)$

13     set $I_{r+1} \leftarrow I_r \setminus E_r$ ;

14     **for** $j = 0, \ldots, k$ **do**

15       $m_j^{(r+1)} \leftarrow m_j - \sum_{i \in [j]} \left| C_i^{(r+1)} \right|$

16   **return** $\{C_1^{(R)}, \ldots, C_k^{(R)}\}$.

---

In this section we show the following theorem.

**Theorem 2.** *For any $R \geq 1$, $\mathtt{SRank}(I, \mathbf{m}, T, R)$ (Algorithm 1) solves the coarse ranking problem with probability at least $1 - 2nR \cdot \exp\left( -\frac{T}{256 \cdot n^{1/R} \cdot R \cdot H^{\langle \mathbf{m} \rangle}(I)} \right)$ using at most $T$ pulls and $R$ rounds.*

*Proof.* The $R$ round cost is clear from the algorithm description. The total number of pulls can be bounded by $\sum_{r=0}^{R-1} n_r \cdot T_{r+1} \leq \sum_{r=0}^{R-1} \frac{n}{n^{r/R}} \cdot \frac{n^{r/R} \cdot T}{nR} \leq T$.

We next prove the correctness of the algorithm and bound the error probability. We first define an event which we will condition on in the rest of the proof.

Let $\pi : \{1, \ldots, n\} \to I$ be the bijection such that $\Delta_{\pi(1)}^{\langle \mathbf{m} \rangle}(I) \leq \Delta_{\pi(2)}^{\langle \mathbf{m} \rangle}(I) \leq \ldots \leq \Delta_{\pi(n)}^{\langle \mathbf{m} \rangle}(I)$. Define event
$$\mathcal{E} \triangleq \left\{ \forall r \in \{0, \ldots, R-1\}, \forall i \in I_r, \left| \hat{\theta}_i^{(r)} - \theta_i \right| < \Delta_{\pi(n_{r+1}+1)}^{\langle \mathbf{m} \rangle}(I)/8 \right\} .$$

The following claim is a simple application of the Chernoff-Hoeffding inequality (Lemma 8). The proof can be found in Appendix B.2.

**Claim 3.** $\Pr[\mathcal{E}] \geq 1 - 2n \cdot R \cdot \exp\left( -\frac{T}{256 \cdot n^{1/R} \cdot R \cdot H^{\langle \mathbf{m} \rangle}(I)} \right)$.

Let $C_1^\star, \ldots, C_k^\star$ be the correct $k$-clustering defined by the input vector $\mathbf{m}$. We show that if $\mathcal{E}$ holds, then $C_j^{(R)} = C_j^\star$. We prove this by induction on round index $r$ with the following induction hypothesis:
$$\forall j \in \{1, \ldots, k\}, \quad C_j^{(r)} \subseteq C_j^\star. \tag{1}$$

Note that (1) holds trivially for $r = 0$. Assuming that (1) holds for $r$, we show it also holds for $r + 1$. The following fact follows immediately from the induction hypothesis.

**Fact 4.** *If the induction hypothesis (1) holds for the $r$-th round, then*

(i) *For any $i \in I_r$, $\Delta_i^{\langle \mathbf{m}^{(r)} \rangle}(I_r) \geq \Delta_i^{\langle \mathbf{m} \rangle}(I)$.*

(ii) *Let $\pi_r : \{1, \ldots, n_r\} \to I_r$ be the bijection such that $\Delta_{\pi_r(1)}^{\langle \mathbf{m} \rangle}(I) \geq \ldots \geq \Delta_{\pi_r(n_r)}^{\langle \mathbf{m} \rangle}(I)$. We have that for any $i \in \{1, \ldots, n_r\}$, $\Delta_{\pi(i)}^{\langle \mathbf{m} \rangle}(I) \leq \Delta_{\pi_r(i)}^{\langle \mathbf{m} \rangle}(I)$.*

Let $\rho_r : \{1, \ldots, n_r\} \to I_r$ be the bijection such that $\theta_{\rho_r(1)} \geq \theta_{\rho_r(2)} \geq \ldots \geq \theta_{\rho_r(n_r)}$. Recall that conditioned on $\mathcal{E}$, we have for any $i \in \{1, \ldots, n_r\}$,

$$\theta_{\rho_r(i)} - \frac{\Delta_{\pi(n_{r+1}+1)}^{\langle \mathbf{m} \rangle}(I)}{8} \leq \hat{\theta}_{\rho_r(i)}^{(r)} \leq \theta_{\rho_r(i)} + \frac{\Delta_{\pi(n_{r+1}+1)}^{\langle \mathbf{m} \rangle}(I)}{8}. \tag{2}$$

The second inequality of (2) implies that $\hat{\theta}_{\sigma_r(i)}^{(r)} \leq \theta_{\rho_r(i)} + \frac{\Delta_{\pi(n_{r+1}+1)}^{\langle \mathbf{m} \rangle}(I)}{8}$, since there are at least $i$ arms with estimated means at most $\theta_{\rho_r(i)} + \frac{\Delta_{\pi(n_{r+1}+1)}^{\langle \mathbf{m} \rangle}(I)}{8}$. Similarly, the first inequality of (2) implies that $\theta_{\rho_r(i)} - \frac{\Delta_{\pi(n_{r+1}+1)}^{\langle \mathbf{m} \rangle}(I)}{8} \leq \hat{\theta}_{\sigma_r(i)}^{(r)}$. We thus have

$$\theta_{\rho_r(i)} - \frac{\Delta_{\pi(n_{r+1}+1)}^{\langle \mathbf{m} \rangle}(I)}{8} \leq \hat{\theta}_{\sigma_r(i)}^{(r)} \leq \theta_{\rho_r(i)} + \frac{\Delta_{\pi(n_{r+1}+1)}^{\langle \mathbf{m} \rangle}(I)}{8}. \tag{3}$$

Inequality (3) establishes the relationship between the empirical order statistics with the real order statistics. As an immediate consequence, we have the following inequality concerning the differences between order statistics.

$$\Delta_i^{\langle \mathbf{m}^{(r)} \rangle}(I_r) - \frac{\Delta_{\pi(n_{r+1}+1)}^{\langle \mathbf{m} \rangle}(I)}{4} \leq \Delta_i^{(r)} \leq \Delta_i^{\langle \mathbf{m}^{(r)} \rangle}(I_r) + \frac{\Delta_{\pi(n_{r+1}+1)}^{\langle \mathbf{m} \rangle}(I)}{4}. \tag{4}$$

Note that the error term $\frac{\Delta_{\pi(n_{r+1}+1)}^{\langle \mathbf{m} \rangle}(I)}{8}$ in (3) is doubled in (4) since we are now considering the difference of two estimates.

Let $E_r^*$ be the set of $(n_r - n_{r+1})$ arms in $I_r$ with the largest gaps $\Delta_i^{\langle \mathbf{m} \rangle}(I)$. By Fact 4 we have that for any $i \in E_r^*$,

$$\Delta_i^{\langle \mathbf{m}^{(r)} \rangle}(I_r) \overset{\text{by Item (i)}}{\geq} \Delta_i^{\langle \mathbf{m} \rangle}(I) \overset{\text{by def. of } E_r^*}{\geq} \Delta_{\pi_r(n_{r+1}+1)}^{\langle \mathbf{m} \rangle}(I) \overset{\text{by Item (ii)}}{\geq} \Delta_{\pi(n_{r+1}+1)}^{\langle \mathbf{m} \rangle}(I). \tag{5}$$

By the first inequality of (4) and (5) we have that for any $i \in E_r^*$,

$$\Delta_i^{(r)} \geq \frac{3\Delta_{\pi(n_{r+1}+1)}^{\langle \mathbf{m} \rangle}(I)}{4}. \tag{6}$$

By the construction of $E_r$ (Line 10 of Algorithm 1) and (6), we have for any $i \in E_r$, $\Delta_i^{(r)} \geq \min_{j \in E_r^\star} \Delta_j^{(r)} \geq \frac{3\Delta_{\pi(n_{r+1}+1)}^{\langle \mathbf{m} \rangle}(I)}{4}$, which, combined with the second inequality of (4), gives that for any $i \in E_r$,

$$\Delta_i^{\langle \mathbf{m}^{(r)} \rangle}(I_r) \geq \Delta_i^{(r)} - \frac{\Delta_{\pi(n_{r+1}+1)}^{\langle \mathbf{m} \rangle}(I)}{4} \geq \frac{\Delta_{\pi(n_{r+1}+1)}^{\langle \mathbf{m} \rangle}(I)}{2}. \tag{7}$$

After these preparation steps we are ready for the induction. We only need to show that for any $j \in [k]$, each arm $i \in C_j^* \cap E_r$ is assigned correctly in the $r$-th round. We prove by contradiction. Suppose that an arm $i \in C_j^* \cap E_r$ is assigned to a cluster $\hat{C}_{j'}^{(r)}$ ($j' \neq j$). We consider two cases: (1) $j' < j$; and (2) $j' > j$. The two cases are symmetric, and thus we only consider the case $j' < j$.

By (7) we have

$$\hat{\theta}_i^{(r)} \geq \hat{\theta}_{\sigma_r(m_{j-1}^{(r)}+1)}^{(r)} + \frac{\Delta_{\pi(n_{r+1}+1)}^{\langle \mathbf{m} \rangle}(I)}{2}. \tag{8}$$

On the other hand, by the assumption that the $i$-th arm was assigned to a cluster $\hat{C}_{j'}^{(r)}$ with $j' < j$, we have $\theta_i \leq \theta_{\rho_r(m_{j-1}^{(r)}+1)}$, which, conditioned on event $\mathcal{E}$, gives

$$\hat{\theta}_i^{(r)} \leq \theta_i + \frac{\Delta_{\pi(n_{r+1}+1)}^{\langle \mathbf{m} \rangle}(I)}{8} \leq \theta_{\rho_r(m_{j-1}^{(r)}+1)} + \frac{\Delta_{\pi(n_{r+1}+1)}^{\langle \mathbf{m} \rangle}(I)}{8}. \tag{9}$$

By (3) we have

$$\hat{\theta}_{\sigma_r(m_{j-1}^{(r)}+1)}^{(r)} \geq \theta_{\rho_r(m_{j-1}^{(r)}+1)} - \frac{\Delta_{\pi(n_{r+1}+1)}^{\langle \mathbf{m} \rangle}(I)}{8}. \tag{10}$$

Combining (9) and (10), we have $\hat{\theta}_i^{(r)} - \hat{\theta}_{\sigma_r(m_{j-1}^{(r)}+1)}^{(r)} \leq \frac{\Delta_{\pi(n_{r+1}+1)}^{\langle \mathbf{m} \rangle}(I)}{4}$, which contradicts (8).

Now we have that for any $r \in \{0, 1, \ldots, R\}$ and for any $j \in [k]$, it holds that $C_j^{(r)} \subseteq C_j^*$. Note that $I_R = \emptyset$ since $n_R = 0$. This means that each input arm has been assigned to some cluster $C \in \{C_1^{(R)}, \ldots, C_k^{(R)}\}$ at the end of the $R$-th round. We thus have $C_j^{(R)} = C_j^*$ for any $j \in [k]$. $\square$

## 3 The Fixed Confidence Case

We now consider the fixed confidence case. Our algorithm is presented in Algorithm 2. The algorithm also follows the SAR framework, but is tailored for the fixed confidence setting. We note that the algorithm for the fixed confidence case is conceptually simpler than the fixed budget case: we do not need to compute the empirical mean gaps of the arms and select those with the maximum gaps to add to the current partially built clusters. We instead gradually build up the clustering by carefully controlling the mean estimation error $\epsilon_r$ at each round $r$ and including those arms whose empirical means are within $\epsilon_r$ of that of closest boundary arm (Line 7-8) to ensure the correctness of the algorithm.

The following theorem summarizes the performance of Algorithm 2. Due to space constraints we delay the proof to Appendix B.3.

**Theorem 5.** *For any error parameter $\delta > 0$, BRank$(I, \mathbf{m}, \delta)$ (Algorithm 2) solves the coarse ranking problem with probability $(1 - \delta)$ using $O\left(H^{\langle \mathbf{m} \rangle}(I) \cdot \log\left(\frac{n}{\delta} \log H^{\langle \mathbf{m} \rangle}(I)\right)\right)$ pulls and $O\left(\log \max_{i \in I} \left\{1/\Delta_i^{\langle \mathbf{m} \rangle}\right\}\right)$ rounds.*

## 4 Lower Bounds

We now show that our algorithmic results in Theorem 5 and Theorem 2 are almost tight. In the fixed budget case, we have the following theorem.

**Theorem 6.** *For any $R \leq \frac{\log n}{\log \log n}$, letting $\mathbf{m} = (0, 1, n)$ and $T = n^{c_T/R} H^{\langle \mathbf{m} \rangle}(I)$ for a sufficiently small universal constant $c_T$, any algorithm that solves the coarse ranking problem in the batched model with input $(I, \mathbf{m}, T)$ with probability at least $0.99$ needs at least $R$ rounds.*

Recall that Algorithm 1 is able to achieve a success probability of $0.99$ using $R$ rounds and $\tilde{O}(n^{1/R} R \cdot H^{\langle \mathbf{m} \rangle}(I))$ pulls (Theorem 2). Theorem 6 thus indicates that Algorithm 1 is almost tight.

In the fixed confidence case, we obtain the following theorem.

**Theorem 7.** *Let $\mathbf{m} = (0, 1, n)$ and $\Delta_{\min} = \min_{i \in I} \Delta_i^{\langle \mathbf{m} \rangle}(I)$. Any algorithm that solves the coarse ranking problem in the batched model with input $(I, \mathbf{m}, 0.01)$ using at most $H^{\langle \mathbf{m} \rangle}(I) \log^{O(1)} n$ pulls needs at least $\Omega\left(\frac{\log(1/\Delta_{\min})}{\log \log(1/\Delta_{\min}) + \log \log n}\right)$ rounds.*

**Algorithm 2:** BRank($I, \mathbf{m}, \delta$)

---

**Input:** a set of arms $I$, cluster boundary vector $\mathbf{m} = (m_0, m_1, \ldots, m_k)$ with
$\quad\quad$ $0 = m_0 < m_1 < \ldots < m_k = n$, and an error parameter $\delta$
**Output:** the coarse ranking ($k$-clustering) of arms in $I$

1 $\;$ Initialize $I_0 \leftarrow I$, $\mathbf{m}^{(0)} \leftarrow \mathbf{m}$, $C_j^{(0)} \leftarrow \emptyset$, $r \leftarrow 0$, and $T_{-1} \leftarrow 0$ ;
2 $\;$ for $r = 0, 1, \ldots$, set $\epsilon_r \leftarrow 2^{-(r+1)}$ and $T_r \leftarrow 8 \cdot \log\left(4n(r+1)^2\delta^{-1}\right)/\epsilon_r^2$ ;
3 $\;$ **while** $I_r \neq \emptyset$ **do**
4 $\quad\quad$ pull each arm in $I_r$ for $T_r - T_{r-1}$ times ;
5 $\quad\quad$ for each $i \in I_r$, let $\hat{\theta}_i^{(r)}$ be the empirical mean of the $i$-th arm after $T_r$ pulls ;
6 $\quad\quad$ let $\sigma_r : \{1, \ldots, |I_r|\} \to I_r$ be the bijection such that $\hat{\theta}_{\sigma_r(1)}^{(r)} \geq \ldots \geq \hat{\theta}_{\sigma_r(|I_r|)}^{(r)}$, for
$\quad\quad\quad$ convenience define $\hat{\theta}_{\sigma_r(0)}^{(r)} = +\infty$, and $\hat{\theta}_{\sigma_r(|I_r|+1)}^{(r)} = -\infty$;
7 $\quad\quad$ **for** $j = 1, \ldots, k$ **do**
8 $\quad\quad\quad$ $C_j^{(r+1)} \leftarrow C_j^{(r)} \bigcup \left\{ i \in I_r : \hat{\theta}_i^{(r)} < \hat{\theta}_{\sigma_r\left(m_{j-1}^{(r)}\right)}^{(r)} - \epsilon_r \; \wedge \; \hat{\theta}_i^{(r)} > \hat{\theta}_{\sigma_r\left(m_j^{(r)}+1\right)}^{(r)} + \epsilon_r \right\}$ ;
9 $\quad\quad$ **for** $j = 0, \ldots, k$ **do**
10 $\quad\quad\quad$ $m_j^{(r+1)} \leftarrow m_j - \sum_{i=1}^{j} \left| C_i^{(r+1)} \right|$ ;
11 $\quad\quad$ $I_{r+1} \leftarrow I \setminus \left( \bigcup_{j=1}^{k} C_j^{(r+1)} \right)$ ;
12 $\quad\quad$ $r \leftarrow r + 1$ ;
13 $\;$ **return** $\{C_1^{(r)}, \ldots, C_k^{(r)}\}$

---

Recall that Algorithm 2 is able to achieve a success probability of 0.99 using $O(\log(1/\Delta_{\min}))$ rounds and $\tilde{O}(H^{\langle \mathbf{m} \rangle}(I))$ pulls (Theorem 5). Theorem 7 thus indicates that Algorithm 2 is almost tight.

Our proofs for Theorem 6 and Theorem 7 make use of the connection between batched algorithms and non-adaptive algorithms in the collaborative learning model, which has recently been proposed for studying multi-agent reinforcement learning [37, 29]. Due to space constraints, we leave the detail of the proofs in Appendix B.4.

## 5 Experiments

In this section, we present the experimental study for our proposed algorithms.

**Algorithms.** As mentioned, our algorithm SRank (Algorithm 1) is the first algorithm for the coarse ranking problem in the fixed budget setting *and* the batched setting. Since top-$m$ arm identifications is a special case of coarse ranking, we first compare SRank with the SAR algorithm [6] for top-$m$ arm identifications. Note that SAR is a fully adaptive algorithm. One would expect it to be a "lower bound" of SRank in terms of the number of pulls. We also use a naive algorithm UNIF for comparison; UNIF just pulls each arm for an equal number of times using one round. One can view UNIF as an "upper bound" of SRank in terms of the number of pulls. We also compare SRank and UNIF in the general coarse ranking setting.

For the fixed confidence case, we compare BRank with the algorithm LUCBRank proposed in [30]. LUCBRank is also a fully adaptive algorithm and can thus be viewed as a lower bound of SRank. We again use UNIF as an upper bound for comparison.

**Datasets.** We test and compare the algorithms on both synthetic and real-world datasets. For synthetic datasets we set the number of arms equal to $n = 500$. In all datasets we assign the $i$-th arm as the Bernoulli distribution with mean $\theta_i$.

- UNIFORM: We have $n = 500$ arms; the mean of the $i$-th arm is set to be $\theta_i = 1 - \frac{i}{n}$.
- NORMAL: We have $n = 500$ arms; the mean of each arm is sampled from a truncated normal distribution with mean 0.5 and standard deviation 0.01, truncated to the range $[0, 1]$.

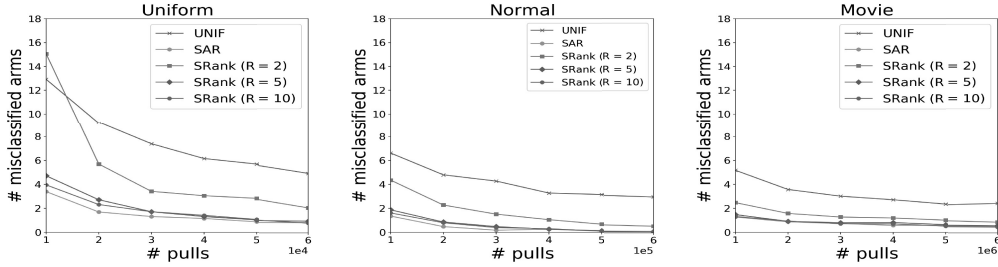

Figure 1: Performance of fixed budget algorithms for top-10 arm identifications

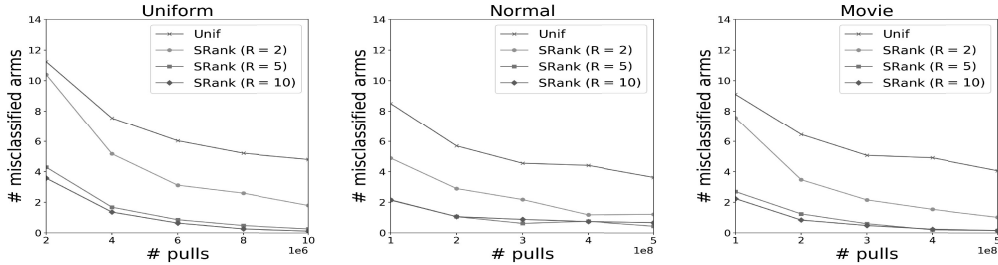

Figure 2: Performance of fixed budget algorithms for coarse ranking

- MOVIE: The MovieLens dataset from [17].[1] We select movies scored by at least $20,000$ users; there are $n = 588$ such movies. For the $i$-th movie we set $\theta_i$ to be the average rating divided by 5 to obtain a value in $[0, 1]$.

**Measurements.** Slightly different from the theoretical guarantees of Theorem 2 and Theorem 5 in which we concern the probability of solving the coarse ranking problem under certain number of pulls and rounds, in our experiments we use the number of misclassified arms as the error measurement for all algorithms. That is, for each arm $i \in [n]$, let $X_i = 1$ if the $i$-th arm is classified into a wrong cluster. For top-$m$ arm identifications, we can think of the set of top-$m$ arms as one cluster and the rest as the other cluster. The error of an algorithm is defined to be $\sum_{i \in [n]} X_i$. We use the number of misclassified arms instead of a Yes/No (correct/incorrect coarse ranking) output due to the following reasons: (1) The number of misclassified arms is a more general measurement than a Yes/No output; it also shows how the algorithms behave under small pull budgets. And (2) the number of misclassified arms is a more stable measurement than a Yes/No output.

All results take an average of 100 runs.

**Computational Environments.** All algorithms are implemented in Kotlin. All experiments are conducted in a PowerEdge R730 server equipped with 2x Intel Xeon CPU E5-2667 v4 3.20GHz. This server has 8-core/16-thread per CPU, and 252GB RAM.

**Results of Fixed Budget Algorithms.** We first compare UNIF, SAR, and SRank(with $R = 2, 5, 10$ rounds respectively) for top-$m$ arm identifications with $m = 10$. The results are presented in Figure 1. Note that under our error measurements, the maximum number of misclassified arms is 20.

We observe that the experimental results are consistent with our theoretical predictions: For SRank in all datasets, there are smooth tradeoffs between the error and the number of rounds of adaptivity. When allowing two rounds ($R = 2$) SRank already outperforms UNIF. When setting $R = 5$ the performance of SRank is almost the same as that of SAR.

We further compare SRank(with $R = 2, 5, 10$) with UNIF on the general coarse ranking problem. We set the number of clusters $k = 5$ and $\mathbf{m} = (m_0, m_2, \ldots, m_5)$ with $m_i = \lfloor in/5 \rfloor$. The results are presented in Figure 2. It is clear that using a few more rounds, SRank significantly outperforms UNIF.

**Results of Fixed Confidence Algorithms.** We compare `BRank` with `LUCBRank` in the fixed confidence setting. We again set the number of clusters $k = 5$ and $\mathbf{m} = (m_0, m_2, \dots, m_5)$ with $m_i = \lfloor in/5 \rfloor$. We set the error parameter as $\delta = 0.01$.

Our results are presented in Figure 3. Recall that `LUCBRank` is a fully adaptive algorithm; given a pull budget $T$ it needs at least $T/10$ policy changes (rounds) which is huge. It is thus not surprising that `LUCBRank` slightly outperforms `BRank` given the same number of pulls. But one can see that the gap between `LUCBRank` and `BRank` is small and `BRank` only needs a few rounds (less than 10 in all cases). We note that some points for `LUCBRank` are not plotted because `LUCBRank` is computational expensive; it cannot return any result within 48 hours in our computational environments.

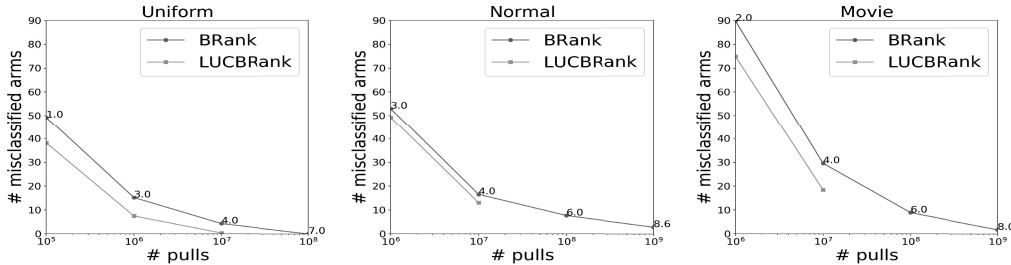

Figure 3: Performance of `BRank` and `LUCBRank` for coarse ranking. The numbers on the curve for `BRank` denote the average number of rounds that `BRank` use, while for `LUCBRank` the number of rounds is at least $T/10$ where $T$ is the pull budget.

# 6  Conclusion

In this paper we have studied the problem of coarse ranking in multi-armed bandit settings, and provided almost optimal algorithms in the batched model. A remaining question in this line of research is to prove almost matching lower bounds for a general boundary vector $\mathbf{m}$; our current lower bounds only hold for a specific vector $\mathbf{m} = (0, 1, n)$.

## Broader Impact

This work can help with ranking objects in various settings such as recommendation systems, peer grading in massive open online courses and paid crowdsourcing platforms. Our ranking algorithms are based on *unbiased* mathematical mechanisms, and is fair to everyone in this regards.

## Acknowledgments

Nikolai Karpov and Qin Zhang are supported in part by NSF CCF-1844234 and CCF-2006591.

## Footnotes

[1]https://grouplens.org/datasets/movielens/25m/

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
