[Supplementary Material]

# Batched Coarse Ranking in Multi-Armed Bandits
## Appendix

## A  Probability Tools

**Lemma 8** (Chernoff-Hoeffding Inequality). *Let $X_1, \ldots, X_n \in [0, d]$ be independent random variables and $X = \sum\limits_{i=1}^{n} X_i$. Then*

$$\Pr[X > \mathbb{E}[X] + t] \leq \exp\left(-\frac{2t^2}{nd^2}\right) \quad and \quad \Pr[X < \mathbb{E}[X] - t] \leq \exp\left(-\frac{2t^2}{nd^2}\right).$$

*Moreover, if $X_1, \ldots, X_n \in [0, 1]$ and $\mu_L \leq \mathbb{E}[X] \leq \mu_H$, then we also have for every $\delta \in [0, 1]$,*

$$\Pr\left[X \geq (1 + \delta)\mu_H\right] \leq \exp\left(-\frac{\delta^2 \mu_H}{3}\right) \quad and \quad \Pr\left[X \leq (1 - \delta)\mu_L\right] \leq \exp\left(-\frac{\delta^2 \mu_L}{3}\right).$$

## B  Missing Proofs

### B.1  Proof for Proposition 1

Let $a_1, \ldots, a_n$ be a sequence of numbers such that $a_1 \geq a_2 \geq \ldots \geq a_n > 0$. The first inequality follows from the simple observation that

$$i \cdot a_i \leq \sum_{j \in [i]} a_j \leq \sum_{j \in [n]} a_j.$$

The second inequality is due to the fact that

$$\sum_{i \in [n]} a_i = \sum_{i \in [n]} \left(\frac{1}{i} \cdot i \cdot a_i\right) \leq \sum_{i \in [n]} \left(\frac{1}{i} \cdot \max_{i \in [n]}\{i \cdot a_i\}\right) \leq \log(2n) \cdot \max_{i \in [n]}\{i \cdot a_i\}.$$

### B.2  Proof for Claim 3

By Chernoff-Hoeffding inequality (Lemma 8) and a union bound we have

$$
\begin{aligned}
\Pr[\bar{\mathcal{E}}] &\leq \sum_{i \in I} \sum_{r=0}^{R-1} \Pr\left[|\hat{\theta}_i^{(r)} - \theta_i| > \Delta_{\pi(n_{r+1}+1)}^{\langle \mathbf{m} \rangle}(I)/8\right] \\
&\leq \sum_{i \in I} \sum_{r=0}^{R-1} 2\exp\left(-T_{r+1} \cdot \left(\Delta_{\pi(n_{r+1}+1)}^{\langle \mathbf{m} \rangle}(I)/8\right)^2\right) \\
&\leq 2n \cdot R \cdot \exp\left(-\frac{T}{256 \cdot n^{1/R} \cdot R \cdot H^{\langle \mathbf{m} \rangle}(I)}\right),
\end{aligned}
$$

where the last inequality follows from Proposition 1.

### B.3  Proof of Theorem 5

We first define the following event which we will condition on in the rest of the proof.

$$\mathcal{G} \triangleq \left\{\forall r = 0, 1, \ldots, \forall i \in I_r, \left|\hat{\theta}_i^{(r)} - \theta_i\right| \leq \epsilon_r/4\right\}.$$

**Claim 9.** $\Pr[\mathcal{G}] \geq 1 - \delta$.

*Proof of Claim 9.* By Chernoff-Hoeffding inequality and a union bound we have

$$
\begin{aligned}
\Pr[\bar{\mathcal{G}}] &\leq \sum_{r=0}^{\infty} \sum_{i \in I_r} \Pr\left[\left|\hat{\theta}_i^{(r)} - \theta_i\right| > \frac{\epsilon_r}{4}\right] \\
&\leq \sum_{r=0}^{\infty} \sum_{i \in I_r} 2\exp\left(-\frac{\epsilon_r^2}{8} \cdot T_r\right) \\
&\leq \sum_{r=0}^{\infty} n \cdot \frac{\delta}{2n(r+1)^2} \leq \frac{\delta \pi^2}{12} \leq \delta.
\end{aligned}
$$

$\square$

Let $\rho : [n] \to I$ be the bijection such that $\theta_{\rho(1)} \geq \theta_{\rho(2)} \geq \ldots \geq \theta_{\rho(n)}$, and for convenience $\theta_{\rho(0)} = +\infty$, $\theta_{\rho(n+1)} = -\infty$. Let $\{C_1^*, \ldots, C_k^*\}$ be the correct $k$-clustering of $I$. We prove the correctness of the algorithm by induction on the round index $r$ using the following induction hypothesis:

1. $\forall j \in \{1, \ldots, k\}, \left|C_j^\star \cap I_r\right| = m_j^{(r)} - m_{j-1}^{(r)}$;

2. $\forall j \in \{1, \ldots, k\}, C_j^{(r)} \subseteq C_j^\star$;

3. $\{i \in I \mid \Delta_i^{\langle \mathbf{m} \rangle}(I) \geq 4\epsilon_r\} \cap I_{r+1} = \emptyset$.

It is easy to see that all items hold trivially for $r = 0$. Assuming that they hold for $r$, we will show that they also hold for $r+1$.

Let $\rho_r : \{1, \ldots, n_r\} \to I_r$ be the bijection such that $\theta_{\rho_r(1)} \geq \theta_{\rho_r(2)} \geq \ldots \geq \theta_{\rho_r(n_r)}$, and for convenience $\theta_{\rho_r(0)} = +\infty$, $\theta_{\rho_r(n_r+1)} = -\infty$. The first and second items of the induction hypothesis imply the following:

$$
\forall j \in [k], \quad \theta_{\rho_r\left(m_j^{(r)}\right)} \geq \theta_{\rho(m_j)} \quad \text{and} \quad \theta_{\rho_r\left(m_{j-1}^{(r)}+1\right)} \leq \theta_{\rho(m_{j-1}+1)}. \tag{11}
$$

We have the following relationship between the empirical order statistics with the real order statistics.

**Claim 10.**
$$
\forall i \in [n_r], \quad \theta_{\rho_r(i)} - \frac{\epsilon_r}{4} \leq \hat{\theta}_{\sigma_r(i)}^{(r)} \leq \theta_{\rho_r(i)} + \frac{\epsilon_r}{4}. \tag{12}
$$

*Proof of Claim 10.* Conditioned on event $\mathcal{G}$ we have that for any $i \in [n_r]$ and $j \leq i$,

$$
\hat{\theta}_{\rho_r(j)}^{(r)} \geq \theta_{\rho_r(j)} - \frac{\epsilon_r}{4} \geq \theta_{\rho_r(i)} - \frac{\epsilon_r}{4}, \tag{13}
$$

which means that there are at least $i$ arms with estimated means more than $\theta_{\rho_r(i)} - \frac{\epsilon_r}{4}$. Consequently we have $\theta_{\rho_r(i)} - \frac{\epsilon_r}{4} \leq \hat{\theta}_{\sigma_r(i)}^{(r)}$. The other half of (12) can be proved symmetrically. $\square$

We now show that each $i \in C_j^{(r+1)} \setminus C_j^{(r)}$ belongs to $C_j^\star$, which implies the first two items in the induction hypothesis.

For each $i \in C_j^{(r+1)} \setminus C_j^{(r)}$, we have

$$
\hat{\theta}_{\sigma_r\left(m_{j-1}^{(r)}\right)}^{(r)} - \epsilon_r \geq \hat{\theta}_i^{(r)} \geq \hat{\theta}_{\sigma_r\left(m_j^{(r)}+1\right)}^{(r)} + \epsilon_r. \tag{14}
$$

Combining (14), (12), and event $\mathcal{G}$ we have

$$
\theta_i \geq \hat{\theta}_i^{(r)} - \frac{\epsilon_r}{4} \geq \hat{\theta}_{\sigma_r\left(m_j^{(r)}+1\right)}^{(r)} + \frac{3\epsilon_r}{4} \geq \theta_{\rho_r\left(m_j^{(r)}+1\right)} + \frac{\epsilon_r}{2}, \tag{15}
$$

and

$$
\theta_i \leq \hat{\theta}_i^{(r)} + \frac{\epsilon_r}{4} \leq \hat{\theta}_{\sigma_r\left(m_{j-1}^{(r)}\right)}^{(r)} - \frac{3\epsilon_r}{4} \leq \theta_{\rho_r\left(m_{j-1}^{(r)}\right)} - \frac{\epsilon_r}{2}. \tag{16}
$$

By (15) and (16), we have $\theta_i \in \left[\theta_{\rho_r\left(m_j^{(r)}+1\right)} + \frac{\epsilon_r}{2}, \theta_{\rho_r\left(m_{j-1}^{(r)}\right)} - \frac{\epsilon_r}{2}\right]$, which implies

$$\theta_i \in \left[\theta_{\rho_r\left(m_j^{(r)}\right)}, \theta_{\rho_r\left(m_{j-1}^{(r)}+1\right)}\right]. \tag{17}$$

Combing (17) and (11) we have $\theta_i \in \left[\theta_{\rho(m_j)}, \theta_{\rho(m_{j-1}+1)}\right]$, which means that $i \in C_j^*$. We thus have $C_j^{(r+1)} \setminus C_j^{(r)} \subseteq C_j^*$, which, combined with induction hypothesis $C_j^{(r)} \subseteq C_j^*$, gives $C_j^{(r+1)} \subseteq C_j^*$. We next handle the third item in the induction hypothesis.

For any arm $i \in C_j^*$, if $i \in I_r$ and $\Delta_i^{\langle \mathbf{m} \rangle}(I) \geq 4\epsilon_r$ then we have

$$\begin{aligned}
\hat{\theta}_i^{(r)} &\overset{\mathcal{G}}{\geq} \theta_i - \frac{\epsilon_r}{4} \\
&\geq \Delta_i^{\langle \mathbf{m} \rangle}(I) + \theta_{\rho(m_j+1)} - \frac{\epsilon_r}{4} \quad (\text{since } i \in C_j^*) \\
&\overset{(11)}{\geq} \Delta_i^{\langle \mathbf{m} \rangle}(I) + \theta_{\rho_r(m_j^{(r)}+1)} - \frac{\epsilon_r}{4} \\
&\overset{(14)}{\geq} \Delta_i^{\langle \mathbf{m} \rangle}(I) + \hat{\theta}_{\sigma_r\left(m_j^{(r)}+1\right)}^{(r)} - \frac{\epsilon_r}{2} \\
&\geq \hat{\theta}_{\sigma_r\left(m_j^{(r)}+1\right)}^{(r)} + 2\epsilon_r.
\end{aligned}$$

Symmetrically,

$$\begin{aligned}
\hat{\theta}_i^{(r)} &\leq \theta_i + \frac{\epsilon_r}{4} \leq -\Delta_i^{\langle \mathbf{m} \rangle}(I) + \theta_{\rho(m_{j-1})} + \frac{\epsilon_r}{4} \\
&\leq -\Delta_i^{\langle \mathbf{m} \rangle}(I) + \theta_{\rho_r\left(m_{j-1}^{(r)}\right)} + \frac{\epsilon_r}{4} \\
&\leq -\Delta_i^{\langle \mathbf{m} \rangle}(I) + \hat{\theta}_{\sigma_r\left(m_{j-1}^{(r)}\right)}^{(r)} + \frac{\epsilon_r}{2} \\
&\leq \hat{\theta}_{\sigma_r\left(m_{j-1}^{(r)}\right)}^{(r)} - 2\epsilon_r.
\end{aligned}$$

Hence, the $i$-th arm should be added to the set $C_j^{(r+1)}$ in the $r$-th iteration and will not occur in $I_{r+1}$.

Now we are ready to prove the correctness of Algorithm 2 and analyze the number of pulls and rounds. By the definition of $\epsilon_r$ and the third item of the induction hypothesis we have that $I_r = \emptyset$ for all

$$r > r_0 = \left\lceil \log\left(4/\min_{i \in I}\left\{\Delta_i^{\langle \mathbf{m} \rangle}(I)\right\}\right)\right\rceil.$$

Therefore the round complexity of the algorithm is bounded by $r_0$. By the second item of the induction hypothesis we have that after $r_0$ rounds the algorithm returns the correct $k$-clustering.

Let $r(i) \triangleq \min\{r \mid \epsilon_r \leq \Delta_i^{\langle \mathbf{m} \rangle}/4\}$. By the third item of the induction hypothesis we know that the $i$-th arm does not occur in any set $I_r$ for any $r > r(i)$, which indicates that the number of pulls of the $i$-th arm is bounded by $T_{r(i)}$. From the definitions of $r(i)$ and $\epsilon_r$, it is clear that $\epsilon_{r(i)} \in [\Delta_i^{\langle \mathbf{m} \rangle}(I)/8, \Delta_i^{\langle \mathbf{m} \rangle}(I)/4]$, which gives

$$T_{r(i)} \leq \frac{512}{\left(\Delta_i^{\langle \mathbf{m} \rangle}(I)\right)^2} \cdot \log\left(\frac{16n \cdot r_0^2}{\delta}\right).$$

Consequently, the total number of pulls is bounded by

$$\sum_{i \in I} T_{r(i)} = O\left(H^{\langle \mathbf{m} \rangle}(I) \log\left(\frac{n}{\delta} \log H^{\langle \mathbf{m} \rangle}(I)\right)\right).$$

### B.4 Proofs of Theorem 6 and Theorem 7

We start by introducing the collaborative learning model [37, 29] and establishing its connection with the batched model. In the collaborative learning model, we have $K$ agents who want to solve a MAB problem together. The learning process is partitioned into rounds. In each round, each of the $K$ agents pulls a (multi)set of arms sequentially. The agents communicate (only) at the end of each round. At the end of the final round (before any communication), all agents should agree on the same output. We assume that for each agent, each pull takes unit time. In this model, we want to minimize the number of rounds $R$ and the total running time $T = \sum_{r=1}^{R} t_r$, where $t_r$ is the maximum number of pulls made among agents in the $r$-th round.

The non-adaptive algorithms for the collaborative learning model is a restricted class of algorithms for which at the beginning of each round, each agent needs to determine the number and the set of arms that it will pull in this round.

It is not hard to see that any algorithm in the batched model can be transformed into a non-adaptive algorithm in the collaborative learning model in the following manner: we can evenly distribute the $T_r$ arm pulls in the $r$-th round in the batched model to the $K$ agents in the collaborative learning model. The running time of the $r$-th round in the collaborative learning model is bounded by $\lceil T_r/K \rceil \leq T_r/K + 1$. We have the following observation. We say an algorithm is $\delta$-error if it succeeds with probability at least $1 - \delta$.

**Observation 11.** *Any $\delta$-error algorithm for coarse ranking in the batched model that uses $T$ pulls and $R$ rounds can be transformed to a $\delta$-error non-adaptive algorithm for coarse ranking in the collaborative learning model that uses at most $R + 1$ rounds and $\frac{T}{K} + R$ time.*

**Proof of Theorem 6.** In [37] the following theorem is shown for the best arm identification problem in MAB in the fixed budget case. Recall that best arm identification is a special case of coarse ranking in which we set $\mathbf{m} = (0, 1, n)$.

**Theorem 12** ([37]). *For any $\alpha \in [1, n^{0.2}]$, letting $\mathbf{m} = (0, 1, n)$ and $T = c_T \cdot \frac{\alpha H^{\langle \mathbf{m} \rangle}(I)}{K}$ for a sufficiently small universal constant $c_T$, any non-adaptive $0.01$-error algorithm for the coarse ranking problem with input parameter $(I, \mathbf{m}, T)$ in the collaborative learning model with $K$ agents needs at least $\Omega \left( \frac{\log n}{\log \log n + \log \alpha} \right)$ rounds.*

Theorem 6 is a direct consequence of Theorem 12 and Observation 11.

**Proof of Theorem 7.** In [37] the following theorem is shown for best arm identification in MAB in the fixed confidence case.

**Theorem 13** ([37]). *Let $\mathbf{m} = (0, 1, n)$ and $\Delta_{\min} = \min_{i \in I} \Delta_i^{\langle \mathbf{m} \rangle}$. Any non-adaptive algorithm that solves the coarse ranking problem with input parameter $(I, \mathbf{m}, 0.01)$ using at most $H^{\langle \mathbf{m} \rangle}(I) \log^{O(1)} n$ arm pulls in the collaborative learning model with $K$ agents needs at least $\Omega \left( \frac{\log(1/\Delta_{\min})}{\log \log(1/\Delta_{\min}) + \log \log n} \right)$ rounds.*

Theorem 7 is a direct consequence of Theorem 13 and Observation 11.