[Reviews · NeurIPS 2020]

Review 1

Summary and Contributions: This paper considers the problem of coarse ranking in multi armed bandits which is a generalization of the top-k arm identification problem. GIven (m_1, \cdots, m_k), the problem is to partition arms into clusters such that cluster i has size m_i and all arms in cluster i are better than all arms in cluster j for i<j. The goal is to minimize the number of arm pulls as well as the number of rounds of interaction with the feedback generating oracle. The paper considers two different settings: (1) a fixed budget setting where there is a fixed budget T for number of arm pulls and a budget R for number of rounds of interaction and the algorithm needs to maximize the probability of correctness; (2) a fixed confidence setting where the algorithm needs to output the correct coarse ranking with a confidence of at least \delta, while minimizing the number of pulls and rounds. The paper gives two algorithms, one for each of these settings. The paper also gives theoretical guarantees on the performance of these algorithms, giving bounds for the number of pulls/rounds and the probability of correctness. Lower bounds for both these settings are also discussed. The paper also includes an experimental evaluation of the proposed algorithms against several baselines, giving validation to the theoretical findings. ********* Post-Rebuttal ************ After reading the authors response I have decided to keep my score unchanged.

Strengths: The main contribution of the paper are the upper bound results. For the fixed budget setting the paper effectively shows that one can reduce the number of (active) arms by a factor of n^{1/R} in each round and increase the budget for each arm accordingly. In doing so one will incur an additional factor of n^{1/R} as compared to the best fully adaptive algorithm. Overall, I think that the paper provides a clean set of results for an important problem. However, the results are not that surprising given the amount of existing work in this field (see below).

Weaknesses: 1. There have been several results on the problems of batched top-k ranking and fully adaptive coarse ranking in recent years. From that point of view the results in this paper are not particularly surprising. Even the idea that one can reduce the size of active arm set by a factor of n^{1/R} has appeared in [37] for the problem of collaborative top-1 ranking. However, the main novelty in this paper seems to be the application of this idea for the problem of coarse ranking using a successive-accepts-and-rejects type algorithm. 2. Also, proving lower bounds for round complexity is the major chuck of work involved in proving results for batched ranking problems. However, this paper exploits an easy reduction from the problem of collaborative ranking, and hence, the lower bound results follow as an easy corollary of these collaborative ranking results.

Correctness: To the best of my knowledge, all claims and methods in the paper are correct. The empirical methodology is also correct.

Clarity: The paper is well written.

Relation to Prior Work: The prior work has been discussed.

Reproducibility: Yes

Additional Feedback: 1. While reading the paper I found that it was easy for me to get lost in the notation, particularly in the proof of Theorem 2. It would be easier if the authors simplify the notation a bit, or perhaps give a proof overview before giving the complete proof. 2. Another setting of interest might be a hybrid of fixed confidence and budget setting, where a fixed confidence \delta for correctness and fixed budget R for the number of rounds are given, and the goal is to minimize the number of pulls.


Review 2

Summary and Contributions: This paper studies coarse ranking in the limited adaptivity / batched setting. In this setting, the learner is allowed a limited number of policy changes. The paper studies both the fixed budget and fixed confidence setting. They propose and analyze algorithms in these settings, and show that their algorithms are tight. == After rebuttal == I have read through the authors' rebuttal and maintain my rating.

Strengths: This paper is very relevant to the bandits and broader ML community, as it expands the scope of the coarse ranking setting. The paper proposes and theoretically analyzes two algorithms, and shows that they are tight. It also studies their empirical performance, and shows that the loss as compared to the fully adaptive setting is small. The paper is well-written.

Weaknesses: The fixed confidence fully adaptive setting of this problem is well-studied. I did not find any discussion on how the bounds with limited adaptivity compare to the fully adaptive setting. In algorithm 1, R is also an input to the algorithm. Second, is there a typo in line 12? Cj^{r+1} seems equal to C_j^r. Line 161 has a typo: tailed -> tailored. The authors should also release the code of their algorithm for reproducibility.

Correctness: Yes

Clarity: Yes

Relation to Prior Work: Yes

Reproducibility: Yes

Additional Feedback:


Review 3

Summary and Contributions: This paper addresses the task of finding as quickly as possible a (predefined) partition of the available arms in a classical stochastic Multi-armed bandit problem. This task, called the coarse ranking problem, is a generalization of the best arm and top-m identification problem in the latter problem scenario. The authors investigate this coarse ranking problem in the batched learning variant, where the learning algorithm should have preferably only a small number of changes regarding the selection of arms (counted by rounds). Moreover, for each conceivable target identification setting, i.e., the fixed budget and the fixed confidence setting, the authors propose a learning algorithm, respectively. Both algorithms adjust the ideas underlying the non-batch learning algorithm SAR, used for the more specific top-m arm identification, to the corresponding setting as well as the batched learning variant. Thorough theoretical analyses are provided for both algorithms and corresponding lower bounds are derived, which show that both algorithms are almost optimal with respect to their corresponding target criterion. Finally, the theoretical considerations are complemented by an experimental study on synthetic datasets as well one real-world dataset revealing satisfactory results.

Strengths: The paper is the first work which investigates the coarse ranking problem in the batched learning variant, which has gained much research interest in the recent past years in various related problem scenarios. Both the fixed budget and the fixed confidence setting are covered by the paper and for each setting the suggested algorithm is proven to be nearly optimal. Although both algorithms are inspired by the non-batch learning algorithm SAR, different technical methodologies are necessary to derive the theoretical results. Finally, it is worth mentioning that the simulation results are in favor of the suggested algorithms.

Weaknesses: Because of the reduction from the batched coarse ranking scenario to the collaborative learning scenario the deviation of the lower bounds is straightforward, so that the theoretical contribution regarding the lower bounds is strictly speaking "donated" by others. On the other hand, one might argue here that one needs to make the connection to the collaborative learning scenario in the first place. As the paper is very theoretical, there is at some points a lack of vivid descriptions of the deviations. However, this is in my opinion a rather minor weakness, as it not unusual that rigorous theoretical analyses in the realm of multi-armed bandits can become rather technical and people interested in these papers are usually accustomed to much mathematical formalism. Another, rather minor, weakness is that there is no conclusion at the end of the paper, where possible paths for future work based on the findings of the paper are discussed.

Correctness: The provided proofs are sound to a large extent. There is only one soft spot, which might be due to a typo or even a misunderstanding of mine and the authors might clarify this in their rebuttal. Namely the deviation in Equation (5), where the definition of E_r^* is used, seems to be flawed, as E_r^* is defined for the population gaps based on the round dependent boundary vector m^(r) as well as the active arm set I_r. However, this seems to be insufficient to infer the second inequality or am I overlooking something? Moreover, the empirical study is well-conducted and supporting the theoretical deliberations before.

Clarity: Although the paper is rather technical, the paper is written in a comprehensible way, as the notation is well thought out (which itself can be a challenging task) and there are only a few typos (please see the detailed comments). There is merely one passage in the experimental study which did not become entirely clear to me (please see the detailed comments). However, I think it would do not harm to explain a bit more the principle underlying the algorithms such as describing the purpose of I_r or m^(r).

Relation to Prior Work: Yes, since the authors provide a paragraph on related work, which explains the difference of their considered setting to previous ones. In addition, the differences are also described to some extent in the experimental study as well as already in the introduction.

Reproducibility: Yes

Additional Feedback: =================== Post rebuttal ====================== After reading the other reviews as well as the author's response, I do not see any reason to change my initial score.


Review 4

Summary and Contributions: This paper studies coarse ranking in multi-armed bandits with bounded number of rounds. They give algorithms in two different variants: the fixed budget model and the fixed confidence model. They also give lower bounds --After feedback-- Thanks for the feedback. I agree with you that the main contribution is the two algorithms. But I would hope you can adjust your wording about tightness and how people should interpret the lower bound results. In the feedback, you said "our upper bounds cannot be improved by more than a logarithmic factor even in the special case". But if I understand your results correctly, you should change the word "even" to "only" in this sentence. Essentially you lower bound only shows that your algorithm is tight on instances in which the problem degenerates to best arm identification. So one can achieve the same level of tightness by running a good best arm identification algorithm when the problem degenerates and an arbitrary coarse ranking algorithm when the problem does not degenerate. And therefore, it's very unclear whether there are algorithms that work much better than the paper's algorithms for coarse ranking.

Strengths: Coarse ranking with bounded interaction is a well-motivated problem. The paper extends the SAR algorithm to work in coarse ranking.

Weaknesses: The lower bound results of the paper only apply to best arm identification. I don’t think they are strong enough to indicate that algorithms in the paper are almost tight for coarse ranking. Otherwise, any good best arm identification algorithms would also be considered tight for coarse ranking.

Correctness: The presented results look correct to me.

Clarity: The paper’s presentation is clear.

Relation to Prior Work: The paper clearly discusses the relation to prior work. One minor comment: Line 51: when you say that these two variants are standard, it might be good to provide citations. I don’t see them very often and most MAB results are about regrets.

Reproducibility: Yes

Additional Feedback:

[Author Response · NeurIPS 2020]

We thank reviewers for their constructive comments on how to improve our paper.

**Common Questions**:

*Regarding the lower bound:* The main contributions of this paper are the two algorithms for the coarse ranking problem.
Our lower bounds are used to *complement* the upper bounds: our upper bounds cannot be improved by more than a
logarithmic factor even in the special case (recall that when $\mathbf{m} = (0, 1, n)$, the coarse ranking problem degenerates to
the best arm identification problem). The lower bound for a general parameter vector $\mathbf{m}$ is an interesting open question
for future study.

*Regarding the clarity:* In the next version of this paper, we will add a table of notations in the introduction and a
conclusion. We will also give more intuition and technical overviews for the proofs.

Below are our responses to individual reviewers.

**Reviewer #1:** We thank the reviewer for the comments on our paper. We have already addressed the comments on the
lower bounds and the presentation in "Common Questions".

**Reviewer #2:** We thank the reviewer for the comments on our paper. Regarding the comparison between our batched
algorithm and the fully adaptive algorithm (LUCBRanking) in [30] in the fixed confidence setting, we would like to
mention two items. First, as briefly mentioned in Line 83 of the submission, it seems difficult to adapt UCB-type
algorithms to the batched setting, since the arm pulls in UCB-type algorithms are inherently sequential. Second, the
complexity measure (instance complexity) in [30] is different from ours, and thus the sample complexities of the two
algorithms are not directly comparable. We will add these discussions to the next version of this paper.

In Algorithm 1, yes, the reviewer is right that $R$ should be part of the input.

In Line 12, we believe the current writing is correct. First recall that $I_r = I \setminus \left( \bigcup_{j=1}^{k} C_j^{(r)} \right)$ and $I_r = \hat{C}_1^{(r)} \cup \ldots \cup \hat{C}_k^{(r)}$.

At the $r$-th round we partition $E_r$ into $k$ subsets $\left\{ E_r \cap \hat{C}_j^{(r)} \right\}_{j=1}^{k}$. According to Line 12 in Algorithm 1, $C_j^{(r+1)}$ is a

superset of $C_j^{(r)}$, and $C_j^{(r+1)} \setminus C_j^{(r)} = E_r \cap \hat{C}_j^{(r)}$. Thus in the general case, $C_j^{(r+1)}$ is *not* equal to $C_j^{(r)}$.

Regarding the code of algorithms, the implementation of our algorithms has already been included in the supplementary
materials of our submission (see the file `code-2904.zip`).

**Reviewer #3:** We thank the reviewer for the detailed feedback on our paper. We will address all the comments on the
presentation in the next version.

In "Common Questions", we have already discussed the functionality of our lower bound results and some writing
improvements that we will conduct in the next version.

On the comment about correctness, we thank the reviewer for pointing out this typo. The definition for $E_r^*$ should be:
"*Let $E_r^*$ be the set of $(n_r - n_{r+1})$ arms in $I_r$ with the largest gaps $\Delta_i^{\langle \mathbf{m} \rangle}(I)$*".

Regarding LUCBRank in the experimental studies, for the first question, yes, the reviewer is right. In each round of
adaptivity, LUCBRank makes at most $2k$ pulls. In our experiments we set $k = 5$, and thus at time $T$ the number of
rounds is at least $T/(2k) = T/10$. For the second question, It is not the averaged number of rounds; it is the lower
bound on the number of rounds.

Regarding the $\log(2n)$ factor, yes, the reviewer is right; we don't need this term. Thank you for pointing this out.

**Reviewer #4:** Regarding Line 51, we will add citations in the next version of the paper. For the best arm identification
problem, both the fixed budget variant and the fixed confidence variant were considered in references [15, 28]. The
fixed budget variant of the top-m problem was considered in reference [6].

[Meta-Review · NeurIPS 2020]

The paper considers a coarse ranking problem that partitions arms in to partially ordered sets of known size. It is a natural problem combining ideas from best arm identification and ranking. Both the fixed confidence and fixed budget settings are addressed. Moreover, limited rounds of adaptivity are discussed to study tradeoffs there. Since the literature is vast on related topics, such a thorough treatment of problem settings was appreciated by the reviewers. While a lower bound is presented, some reviewers were concerned about it only matching in special cases. Taken as a whole, the paper merits acceptance.